# The Effectiveness of Pharmacist Interventions in the Management of Patient with Renal Failure: A Systematic Review and Meta-Analysis

**DOI:** 10.3390/ijerph191811170

**Published:** 2022-09-06

**Authors:** Magdalena Jasińska-Stroschein

**Affiliations:** Department of Biopharmacy, Medical University of Łódź, Ul. Muszyńskiego 1, 90-151 Lodz, Poland; magdalena.jasinska-stroschein@umed.lodz.pl

**Keywords:** renal failure, pharmacist-led intervention, blood pressure, adherence, disease management, meta-research

## Abstract

The existing trials have focused on a variety of interventions to improve outcomes in renal failure; however, quantitative evidence comparing the effect of performing multidimensional interventions is scarce. The present paper reviews data from previous randomized controlled trials (RCTs), examining interventions performed for patients with chronic kidney disease (CKD) and transplants by multidisciplinary teams, including pharmacists. Methods: A systematic search with quality assessment was performed using the revised Cochrane Collaboration’s ‘Risk of Bias’ tool. Results and Conclusion: Thirty-three RCTs were included in the review, and the data from nineteen protocols were included in further quantitative analyses. A wide range of outcomes was considered, including those associated with progression of CKD, cardiovascular risk factors, patient adherence, quality of life, prescription of relevant medications, drug-related problems (DRPs), rate of hospitalizations, and death. The heterogeneity between studies was high. Despite low-to-moderate quality of evidence and relatively short follow-up, the findings suggest that multidimensional interventions, taken by pharmacists within multidisciplinary teams, are important for improving some clinical outcomes, such as blood pressure, risk of cardiovascular diseases and renal progression, and they improve non-adherence to medication among individuals with renal failure.

## 1. Introduction

Chronic kidney disease (CKD) is common, with an estimated global prevalence above ten percent. This high rate is also being driven by an ageing population with an increased risk of cardiovascular diseases (CV) [1]. As CKD is associated with multiple comorbidities, including obesity, diabetes, and hypertension [2], forms of lifestyle modification, BP control, and anti-hypertensive medication, such as RAAS inhibitors and lipid modification, are recommended to reduce the risk of renal failure. Strategies for achieving glycemic control are also desirable in patients with diabetes mellitus [1]. Such interventions are particularly important when considering that, despite being asymptomatic, most patients with mild-to-moderate CKD still have a higher risk of CV, anemia, metabolic bone disease, or progression to end-stage renal disease (ESRD) requiring hemodialysis [3].

For example, reaching target blood pressure (BP) goals can dramatically reduce the risks of these complications. Although the rate of BP goal achievement is only around 50% in patients with CKD [4], a recent meta-analysis found that pharmacist interventions focusing on home-based BP telemonitoring can increase the odds ratio for BP goal achievement in this group [5].

Achieving optimal health outcomes in CKD patients, i.e., disease prevention and management, is also hampered by drug-related problems (DRPs), which often lead to hospitalizations and can result in clinical outcomes correlated with decreased quality of life. Interventions focusing on DRPs are varied and cover a broad range of aspects, such as medication reconciliation, dose adjustment, therapeutic indication, or medication adherence. Non-adherence to pharmacological treatment has been associated with an increased risk of poor health, adverse clinical problems, and mortality due to chronic diseases [6]. In turn, the potential benefits of medication adherence include improved survival, enhanced quality of life, and lower risk of hospitalization [7]. This can also be said for ESRD patients receiving hemodialysis; this group demonstrates multiple complications requiring pharmacologic therapy, and ESRD may heighten the risk of unfavorable drug effects. In addition, the group tends to demonstrate worse health-related quality of life compared with the general population [8].

The administration of multiple medications, as well as poor compliance with drug regimens and drug interactions, may increase the risk of DRPs [9]. Clinical pharmacists are, therefore, in a unique position to decrease the appearance rate of DRPs and, thus, improve clinical outcomes and quality of life. Existing trials have focused on a variety of interventions (e.g., psychological support and motivation, pharmacist medication review, BP monitoring), including team-based ones, to improve outcomes in CKD. A recent systematic review has assessed the relative effectiveness of CKD management strategies based on models of care built around different health care providers, e.g., multidisciplinary specialists, nurses, or pharmacists, as well as various outcomes, including renal, cardiovascular, and mortality. Nurse and pharmacist-led care reported improved prescription rates of drugs relevant to CKD, while nurse-led and multidisciplinary specialist care were associated with small improvements in blood pressure control [1]. However, pooled quantitative evidence comparing the effect of such interventions is scarce.

The present survey is an updated review and quantitative analysis of data from randomized controlled trials assessing various interventions performed for CKD and transplant patients by multidisciplinary teams consisting of pharmacists and, perhaps, other health professionals. A wide range of outcomes was considered, including those associated with the progression of CKD, cardiovascular risk factors, DRPs with special attention to patient adherence, quality of life, prescribing of relevant medications, rate of hospitalizations, and death.

## 2. Materials and Methods

The study followed the Cochrane guidelines for conducting systematic reviews. It is reported in accordance with the Preferred Reporting Items for Systematic Review and Meta-analysis (PRISMA) statement.

### 2.1. Search Strategy

The search strategy was run on three data-bases (PubMed: 1966 to 2022, EMBASE: 1974 to 2022 and Ebsco: 1974 to 2022), according to the following search criteria: ‘pharmacist AND (renal failure OR kidney disease OR hemodialysis OR transplantation)’.

### 2.2. Study Selection

The inclusion criteria were as follows: randomized controlled trials (RCTs) that were conducted on (i) individuals aged 18 years and above from any country, (ii) individuals with renal failure associated with acute or chronic kidney disease, hemodialysis, or transplant patients (this was obligatory), (iii) and, optionally, with a comorbid condition, such as diabetes, hypertension, and other cardiovascular disease (CV) as stroke, atrial fibrillation, or coronary artery disease. The model of care must have been capable of delivering an intervention targeted for disease management; baseline characteristics and study outcomes must have been reported for at least a control group (Usual care) and a group that received interventions from a clinical or community pharmacist (Intervention), either alone or in co-operation with other health care providers. Studies that did not meet the population, intervention, control, or outcome criteria were excluded. In order to limit the potential for bias, observational studies, review articles, and conference presentations were excluded.

### 2.3. Data Extraction and Quality Assessment

Standardized data extraction forms were developed. The collected data included setting, population details, and details of the intervention, as well as outcomes of interest and quality assessment items. Quality assessment was carried out using the revised Cochrane Collaboration’s ‘Risk of Bias’ tool (RoB 2) [10]; the highest degree of risk for any of these domains is then used to determine the overall risk of bias. Risk of bias was assessed independently by two reviewers (M.J.-S., D.W.), and any differences were resolved by consensus.

### 2.4. Data Analysis

Where possible, the outcome results were calculated with probability (P) values and 95% confidence interval (CI) for each study. The overall effect size was expressed as mean differential change from baseline: intervention vs. usual care for continuous data, e.g., blood pressure values, which were measured at the beginning and at the end of the study for patients from each group. The odds ratio was calculated for dichotomous data, where a particular study reported an association between exposure (intervention or usual care) and outcome (e.g., proportion of patients with achieved BP goal). A random-effects model was used to compensate for the heterogeneity of studies. Heterogeneity was quantitatively assessed using Cochran’s Q and by the chi-square test (I2). A statistically-significant Q measure (*p* < 0.05) indicated heterogeneity among two or more analyzed subgroups. Duval and Tweedie’s trim and fill adjustment, as well as the Egger test, were used to assess bias across the studies. A two-tailed *p*-value, less than 0.05, was considered statistically significant.

### 2.5. Outcomes

Outcomes of interest included inter alia clinical endpoints (e.g., progression of CKD, eGFR, proteinuria expressed by urine protein(albumin)/creatinine ratio), risk factors (e.g., blood pressure, anemia, hypercholesterolemia, HbA1c, tobacco cessation, overall cardiovascular risk), DRPs with special attention to patient adherence, quality of life, prescribing of relevant medications, rate of hospitalizations, and death.

## 3. Results

### 3.1. Search Results and Description of Included Studies

The search yielded 1413 papers. After adjustment for duplicates, 612 studies remained. Of these, after screening the titles and abstracts, 539 articles were found to be irrelevant to the review question and were discarded. The final corpus comprised 33 papers, as demonstrated in the PRISMA flow diagram (Figure 1) [11]. The total number of study participants is 10,710, ranging between ages 60.9 ± 8.3 (Intervention) and 61.2 ± 8.6 (Usual care).

### 3.2. Quality Assessment of Included Studies

#### 3.2.1. Bias Arising from the Randomization Process

In 54.5% of the included studies, the risk of bias was unclear (i.e., with some concerns). In particular, the process of random sequence generation was not reported (45.4%), the method used to allocate participants was not described, or the allocation itself was not concealed (69.7%). In case of highly-biased studies (13%), the overall methodological quality of evidence was decreased by a lack of baseline characteristics in the study groups, e.g., usual care vs. intervention (3.0%). In some cases, baseline differences between intervention and usual care groups can suggest a problem with the randomization process (9.1%).

#### 3.2.2. Bias Due to Deviations from the Intended Interventions

In the majority of included studies, the participants were aware of the intervention due to its nature. In addition, 42.4% of studies were semi-blind (investigators), e.g., the staff nurse or research assistant was blinded to the intervention. In 10% of studies, the data were not analyzed according to the intention-to-treat principle. In some protocols, no information was provided about blinding (42.4%) or the type of analysis used to estimate the effect of assignment to intervention (45.4%). Another risk of bias was that it was not possible to assess whether participants were analyzed in the wrong intervention group or were excluded from the analysis; this could have had a substantial impact on the result, due to lack of consort flow (15.1% of studies). In total, 50% of protocols were unclear, and 50%-demonstrated high risk of bias.

#### 3.2.3. Bias Due to Missing Outcome Data

The risk of bias was unclear (some concerns) in 15.1% of the included studies, and no information about percentage of missing values was provided in 27.3%. Others were characterized with high proportions of missing data (>10–30%) or demonstrated differences between groups regarding the proportion of missing outcome data (15.1%).

#### 3.2.4. Bias in Measurement of the Outcome

In 15.1% of the included studies, the outcome assessors were aware of the intervention received. This could have had a substantial impact on the result, e.g., where participants reported outcomes (the quality of life, adherence). In some cases (50.0%) it was not known whether the study was blinded and/or whether the assessments were influenced by the knowledge of the intervention. The percentage risk of unclear bias was 39.4%.

#### 3.2.5. Bias in Selection of the Reported Result

In most cases, the result being assessed was likely to have been selected from multiple analyses of the data (12.1%). This mainly concerned the measure of patient adherence. Another point was that, in some protocols, only final values were reported without reference to the baseline characteristics (6.1%), e.g., blood pressure or DRPs were only reported in participants from the usual care and intervention groups at the end of study. The percentage risk of unclear bias was 21.2%, and that of a high risk of bias was 18.2%. Overall, N = 4 (12.1%) studies were judged as low risk of bias, N = 8 (24.2%) as unclear, and N = 21 (63.7%) as high risk. For the studies included in the meta-analysis, N = 1 (5.3%) was judged as low risk of bias, N = 6 (31.6%) as unclear, and N = 12 (63.1%) as high risk. The summary of bias is presented in Figure 2. Further analyses were performed to assess publication bias. Due to the limited amount of data, the analyses only concerned the protocols with blood pressure, i.e., the most commonly reported data. In this case, a fairly symmetrical plot with a non-significant Egger test result indicates the absence of publication bias across the studies, with regard to differential change in SBP and DBP from the baseline. This was not true, however, for odds ratios that described the percentage of patients who achieved target BP (Figure 3).

### 3.3. Study Characteristics

Thirty-three RCTs were identified and included in this review (Table 1). In 26 papers, 100% of patients had renal failure, as defined by each study, or were after kidney transplantation. Among these papers, 8 from 33 studies concerned hemodialysis subjects only. Among 33 studies, 3 described an intervention by community pharmacists, with the patient-oriented activities performed by clinical pharmacists or renal pharmacists in the remaining protocols; in most studies, these pharmacists worked as part of multidisciplinary care teams with nephrologists (or other health care professionals), nurses, and nutritionists. Most frequently, pharmacists assessed patient knowledge and provided patient motivation and education about medications (e.g., goals of therapy, potential side effects, and/or contraindications), disease, lifestyle changes, and nutrition (N = 24 studies, 72.7%).

**Table 1 ijerph-19-11170-t001:** Study characteristics.

Source/ Location	Study Type, Sample Size ¥, Follow up (Months)	Participants (Baseline Percentage of Patients with Renal Failure ¥)	Comorbidities	Key Components of Pharmacist Intervention	Measured Outcomes (Intervention vs. Usual Care)
Anderegg MD et al. 2018 [12] USA	Cluster RCT 227/108 9	HTN (12.3/13.9%)	Arthritis, asthma/COPD, CAD, CKD, depression/anxiety, DM, Dys, HF, PVD, stroke/TIA	Pharmacists conducted MRs and assessed patient knowledge of BP medications, goals of therapy, medication dosages, potential side effects, contraindications, and monitoring; then created an individualized care plan with BP goal and medicine recommendations (1).	primary: SBP reduction at 9th month from baseline (8.64 mmHg; 95% CI −12.8–(−4.49); *p* < 0.001) #; BP control by the JNC−7 or JNC−8 inclusion (OR = 1.97; 95% CI 1.01−3.86; *p* = 0.047) or (OR = 2.16; 95% CI 1.21−3.85; *p* = 0.0102)
Carter BL et al. 2015 [13] USA	Cluster RCT 401/224 9	HTN (47.4/54.0% *; 47.3/54.0% **)	CKD (3a–5) or DM	Pharmacist conducted MRs. The model recommended a telephone calls, structured face−to−face visits, and additional visits if BP remained uncontrolled. Dose adjustments (1).	primary: BP control at 9th month (OR = 1.57; 95% CI 0.99–2.50; *p* = 0.059); secondary: BP control at 12th, 18th and 24th month
Magid DJ et al. 2013 [14] USA	RCT 162/164 6	HTN (46.3/50.9%) ^	CKD (1–3b) or DM	Pharmacist conducted home BP telemonitoring, and reviewed current BP medication regimen, provided counselling on lifestyle changes, educational materials, and adjusted or changed antihypertensive medications as needed, including dosage (1).	primary: the proportion of patients who attained goal BP (<130/80 mmHg for CKD) at 6th month (adjusted OR = 3.84; 95% CI 2.08–7.10; *p* < 0.0001); secondary: change at 6th month from baseline in SBP (−15.4 mmHg; 95% CI; −21.0–(−9.8); *p* < 0.0001) and DBP (−7.3 mmHg; −10.4–(−4.1); *p* < 0.0001); change in antihypertensive medication intensity (70% vs. 25%; *p* < 0.001); hospitalizations (OR = 0.70; 95% CI 0.22–2.24; *p* > 0.05); change in medication intensity score (1.35 ± 1.37 vs. 0.15 ± 0.82; *p* < 0.001); medication usage: CCB (OR = 2.61; 95% CI 1.6−4.18; *p* < 0.0001); ACEI/ARB (OR = 1.59; 95% CI 0.98−2.58; *p* = 0.06); diuretic (OR = 8.80; 95% CI 4.66−16.61; *p* < 0.001) at 6th month.
Margolis KL et al. 2013 [15] USA	RCT 188/182 18	HTN (17.1/14.6%)	CKD (3a–5), DM, obesity	Pharmacist conducted home BP telemonitoring, and reviewed the patient’s relevant history, instructed patients on BP telemonitor system and the individualized home BP goal using the home (1).	primary: BP control to <130/80 mmHg in patients with diabetes or kidney disease) at 6th (27.2%; 95% CI 13.4–40.0; *p* < 0.001) and 12th month (29.6%; 95% CI 13.1–46.0; *p* < 0.002); secondary: change in SBP (−10.70 mmHg; 95% CI −14.90–(7.31); *p* < 0.0001) #; change in DBP (−6.00 mmHg; 95% CI −8.61–(−3.39); *p* < 0.0001) #; patient satisfaction (0.15; 95% CI 0.03–0.28; *p* = 0.014) # at 18th month from baseline, BP control (14.7%; 95% CI 7.0–21.4; *p* < 0.003); medication adherence (OR = 1.50; 95% CI 0.98–2.31; *p* = 0.064) at 18th month; change in HRQoL–SF-12 physical function (0.28; 95% CI 0.15−0.41; *p* = 0.0001) #; SF-12 mental function (1.01; 95% CI 0.85−1.17; *p* < 0.0001) #; % of smoked in last 30 days (−2.0; 95% CI −2.44–(−1.56); *p* < 0.0001) # at 18th month from baseline.
Al Hamarneh YN et al. 2017 [16] Canada	RCT 286/287 3	DM and at least 1 uncontrolled risk factor (40.9/42.2%)	AF, CAD, CKD (3–5), Dys, HF, HTN, MI, PVD, stroke/TIA	MTM: pharmacist assessed patient BP, waist circumference, weight and height measurements, A1C level, lipid profile, kidney function and status, individualized CV risk; provided treatment recommendations, prescription adaptations and/or initiation (1).	primary: change in estimated CV risk from baseline to 3 months after randomization (5.38; 95% CI 4.24−6.52; *p* < 0.001) #; secondary: differences in changes in medication use and dose changes
Al Hamarneh YN et al. 2018 [17] Canada	RCT 147/143 3–6	CKD, stage 3a–5 (100%)	AF, CAD, DM, Dys, HF, HTN, MI, PVD, stroke/TIA	MTM: pharmacist assessed patient BP, waist circumference, weight and height measurements, A1C level, lipid profile, kidney function and status, individualized CV risk; provided treatment recommendations, prescription adaptations and/or initiation (1).	primary: change in estimated CV risk (5.03; 95% CI 3.4−6.65; *p* < 0.001) #; secondary: change in LDL (0.2 mmol/L; 95% CI 0.1−0.4; *p* = 0.004) #; HbA1c (0.7; 95% CI 0.4−0.9%; *p* < 0.001) #; tobacco cessation (19.6%; *p* = 0.04) and SBP (10.50 mmHg; 95% CI 7.40−13.50; *p* < 0.0001) # from baseline to 3 months
Chang AR et al. 2016 [18] USA	Cluster RCT 24/23 12	CKD, stage 3a (100.0%)	CAD, DM, dys, HF, HTN	MTM: pharmacist conducted MRs, ordered lipid and ACR screening tests, and managed BP and lipid therapy. Patients were contacted by telephone and were scheduled for clinic visits with the pharmacist for medication initiation and/or titration (1).	primary: screening for proteinuria by urine ACR or protein/creatinine ratio at 24th month (OR = 2.6; 95% CI 0.5– 4.0; *p* > 0.05); secondary: BP control (OR = 0.9; 95% CI 0.3–3.0; *p* > 0.05); the proportion of patients who were on statins (OR = 0.4; 95% CI 0.12–1.3; *p* > 0.05)
Cooney D et al. 2015 [19] USA	RCT 441/429 12	CKD, stage 3 (100.0%)	CAD, DM, HF, HTN	Multifactorial intervention: a phone−based pharmacist intervention, pharmacist−physician collaboration, patient education (informational pamphlet), and a CKD registry (1).	primary: BP control in subjects with poorly controlled hypertension at baseline (OR = 1.03; 95% CI 0.79−1.35; *p* > 0.05); measurement of PTH during the study (46.9% vs. 16.1%; *p* < 0.001); secondary: number of classes of antihypertensive drugs being prescribed (*p* = 0.02); change in HRQoL (52.0 ± 10.6 vs. 52.1 ± 9.6: *p* > 0.05); mortality (RR = 1.40; 95% CI 0.99−1.99; *p* = 0.06); medication adherence (6.8 ± 1.2 vs. 6.7 ± 1.2; *p* > 0.05).
Peralta CA et al. 2020 [20] USA	RCT 616/604 14	CKD, stage 3a–5 (100.0%)	CAD, CVD, COPD, HF, Dys, malignancy, mental health disorders	An electronic letter to the primary care provider included an opt−in option to refer persons with newly detected CKD to a clinical pharmacist for MRs, drug adjustment, CKD education and counselling (1).	primary: change in SBP (−1.00 mmHg; 95% CI −1.69–(−0.31); *p* < 0.05) #; DBP (−1.00 mmHg; 95% CI −1.35–(−0.65); *p* < 0.05) # at 14th month from baseline; secondary: controlled BP at 14th month (62% vs. 64%; *p* > 0.05); medication usage: ACE/ARB (OR = 1.11; 95% CI 0.88−1.43; *p* > 0.05); diuretics (OR = 0.97; 95% CI 0.75−1.26; *p* > 0.05); statins (OR = 0.83; 95% CI 0.54−1.26; *p* > 0.05); NSAIDs (OR = 0.99; 95% CI 0.76−1.29; *p* > 0.05).
Peralta CA et al. 2020 [21] USA	Cluster RCT 171/188 12	CKD, stage 3 (41.5/52.1%)	CAD, CVD, HF, DM, HTN, Dys	Pharmacist scheduled a follow−up visit by telephone in order to reinforce medication changes proposed by PCP, consulted patients about CKD and conducted MRs (1).	primary: change in SBP (−1.10 mmHg; 95% CI −6.69–4.49; *p* > 0.05); DBP (−0.40 mmHg; 95% CI −6.63–5.83; *p* > 0.05) at 12th month from baseline; secondary: controlled BP at 12th month (OR = 1.02; 95% CI 0.67–1.55; *p* > 0.05); PCP awareness of CKD at study end (OR = 2.57; 95% CI 1.46–4.54; *p* = 0.001); medication usage: ACEI/ARB (OR = 0.77; 95% CI 0.50−1.16; *p* > 0.05); diuretics (OR = 0.69; 95% CI 0.42−1.15; *p* > 0.05)
Santschi V et al. 2011 [22] Canada	Cluster RCT 48/41 6	CKD, stage 3a–5 (100.0%)	CV, DM, Dys, HTN	ProFiL: patients’ clinical pharmacist summaries were sent to the community pharmacists to facilitate the detection of DRPs (list of patients’ health problems, eGFRs and medications). Community pharmacists could consult a nephrology pharmacist, when needed. Patient education (2).	Adjusted mean SBP (−11.60 mmHg; 95% CI −21.30–(−1.90)); *p* = 0.019) # and DBP changes (−2.60 mmHg; 95% CI −8.11–2.91; *p* > 0.05) # at 6th month from baseline. Patients with written recommendations had a greater decrease in mean systolic BP (−11.6 mmHg; *p* = 0.035), and BP was controlled in a higher proportion of them (RR = 2.14; *p* = 0.011); medication usage: diuretic (OR = 1.41; 95% CI 0.76–2.63; *p* > 0.05); ACEI/ARB (OR = 1.39; 95% CI 0.75–2.57; *p* > 0.05); CCB (OR = 1.38; 95% CI 0.71–2.65; *p* > 0.05).
Cypes IN et al. 2021 [23] USA	RCT 95/87 3	CKD, stage 2–5 (70.5/64.4%)		Pharmacist conducted medication recommendations in order to identify and resolve medication-dosing errors and improve collaboration: providers–pharmacists (1).	primary: the number of medications requiring pharmacist intervention (22.1 vs. 19.5%); incorrect CKD staging (48.4% vs. 55.2%); rate of provider response to pharmacist−initiated medication recommendations (*p* > 0.05)
Lalonde L et al. 2017 [24] Canada	Cluster RCT 304/138 12	CKD, stage 2–5 (100.0%)	Anemia, CAD, DM, Dys, HTN	Pharmacists received interactive web−training program, a clinical pharmacist guide (a booklet on CKD drug therapy), a clinical pharmacist summary of their patients with information on their kidney function, and a consultation service with pharmacists working in a CKD clinic (2).	Change in DRPs per patient (−0.32; 95% CI −0.60 –(−0.0063); *p* = 0.04) #; improvements in knowledge (4.5%; 95% CI 1.6–7.4; *p* = 0.0024) and clinical competencies (7.4%; 95% CI 3.5–11.3; *p* = 0.0002) #; uncontrolled BP (−0.20 mmHg; 95% CI −0.41–0.10; *p* > 0.05) #; non−optimal adherence to drug therapy (−0.14; 95% CI −0.48–0.20; *p* > 0.005) #; drugs requiring dose adjustment in CKD (−0.03; 95% CI −0.12–0.06; *p* > 0.05) #; eGFR (2.5 mL/min 1.73 m^2^; 95% CI −1.8–6.7; *p* > 0.05) #; SBP (−1.2 mmHg; 95% CI −4.8–2.5; *p* > 0.05) #; DBP (0.01 mmHg; 95% CI −1.95–1.97; *p* > 0.05) #; LDL−C (20.1; 95% CI 20.2–0.1; *p* > 0.05) #; HbA1c (0.1%; 95% CI −0.2–0.1; *p* > 0.05) # 12th month from baseline
Quintana−Bárcena *p* et al. 2018 [25] Canada	Cluster RCT 304/137 12	CKD, stage 2–5 (99.6/99.5%)	CAD, DM, Dys, HTN	ProFiL: patients’ clinical pharmacist summaries were sent to the community pharmacists for detection of DRPs (list of patients’ health problems, eGFRs and drugs). Community pharmacists could consult a nephrology pharmacist. Patient education (2).	The prevalence of DRPs per patient–mild (0.55 ± 0.98) and moderate (1.04 ± 1.51). An unadjusted change in moderate DRPs (0.32; 95% CI =−0.6–(−0.06); *p* < 0.05) # at 12th month from baseline.
Song YK et al. 2021 [26] South Korea	RCT 50/50 3	CKD, stage 2–5; dialysis (100.0%)	Anemia, DM, Dys, HTN, mineral bone dis, hyperuricemia	DrugTEAM: pharmacists provided MRs, communicated with healthcare professionals, patients. Drugs were documented at discharge and the patient were counselled using educational materials (1).	primary: the number of DRPs per patient at discharge (0.9 ± 1.0 vs. 2.0 ± 1.3; *p* < 0.001); patients with any DRPs at discharge (OR = 0.14; 95% CI 0.04−0.046; *p* = 0.001); secondary: patients with increased MMAS score (OR = 2.46; 95% CI 0.96−6.27; *p* = 0.06)
Thanapongsatorn P et al. 2021 [27] Thailand	RCT 40/38 12	AKI, CKD stage 2–3 (100.0%)	CAD, CVD, CKD, DM, HF, HTN, liver disease, malignancy	Multidisciplinary care team consisted of nephrologists, renal nurses, renal pharmacists, and nutritionists. Renal pharmacist conducted the drug reconciliation, alerted the nephrologists, dealt with dosing errors, drug interactions, or nephrotoxins, provided medication education and adjustment of the medication dosage based on the renal function (1).	primary: feasibility outcomes: rate of 3−d dietary record (100.0% vs. 0.0%; *p* < 0.0001), rate of drug reconciliation (100.0% vs. 0.0%; *p* < 0.0001), and rate of drug alerts (30.0% vs. 0.0%; *p* < 0.0001); secondary: eGFR at 12 months (66.74 ± 30.77 vs. 61.23 ± 35.16 mL/min 1.73 m^2^; *p* > 0.05); urine albumin: creatinine ratio (UACR) (36.83 mg/g; 95% CI 13.39−131.90 vs. 177.70 mg/g; 95% CI 47.12−745.71; *p* = 0.036); BP control (OR = 5.34; 95% CI 1.53−18.70; *p* = 0.0088); CKD progression (OR = 0.48; 95% CI 1.0.08−2.81; *p* > 0.05); mortality (RR = 1.28; 95% CI 0.44–3.74; *p* > 0.05); medication usage: ACEI/ARB use (OR = 1.37; 95% CI 0.51−3.70; *p* > 0.05) at 12th month
Theeranut A et al. 2021 [28] Thailand	RCT 166/168 3	CKD, stage 2−3b (100.0%)	CAD, DM, Dys, gout, HTN	Chumpae model for delaying dialysis in CKD patients. The clinical pharmacist information system provided updated, systematic clinical pharmacist evidence for the patients and care team (1).	primary: change in eGFR (7.61 mL/min 1.73m^2^; 95% CI 5.83−9.39) # at 3th month from baseline; proportion of patients with eGFR decline greater than 4 mL/min/1.73 m^2^ (OR = 0.17; 95% CI 0.1−0.28; *p* < 0.001); difference in CKD stage from baseline (*p* < 0.001)
Tuttle KR et al. 2018 [29] USA	RCT 66/63 3	CKD, stage 2–5 (92.7/97.2%)	DM, HTN	Pharmacist conducted MRs, taken medication action plan and personal medication list, within in−home visits 7 days after hospital discharge. The intervention was augmented by the Chronic Care Model and on the basis of an algorithm for the “5As” (Assessment, Advice, Agreement, Assistance, and Arrangements) process, including medication doses adjustment for kidney function (1).	primary: a composite of first acute care events (hospitalization and emergency department) within the 90−day period after index hospitalization (OR = 1.17; 95% CI 0.60–2.29; *p* > 0.05); secondary: hospitalization (OR = 1.02; 95% CI 0.48–2.15; *p* > 0.05); eGFR (0.01 mL/min 1.73 m^2^; 95% CI −0.96–0.96; *p* > 0.05) #; SBP (−1.00 mmHg; 95% CI −2.36–0.36; *p* > 0.05) #; DBP (1.99 mmHg; 95% CI; 1.21–2.77; *p* < 0.05) #; BP goal < 130/80 mmHg (OR = 1.33; 95% CI 0.65–2.73; *p* > 0.05); BP goal < 140/90 mmHg (OR = 1.19; 95% CI 0.55–2.59; *p* > 0.05); hemoglobin (0.01 g/dL; 95% CI −0.66–0.66; *p* > 0.05) #; HbA1c (−0.2%; 95% CI −0.27–(−0.13); *p* < 0.0001) # at 3rd month from baseline
Wilson FP et al. 2015 [30] USA	RCT 1201/1192 1	AKI (100.0%) CKD (27/26%)	CVD, CKD (1–3b), DM, HF, metastatic disease	Automated, electronic alerts for AKI received by an intern, resident, or nurse practitioner and unit pharmacist (1).	primary: composite of relative maximum change in creatinine, dialysis, and death at 7 days after randomization (*p* > 0.05); median relative change in creatinine concentrations (0.0; IQR 0.0–18.4; vs. 0.6; IQR 0.0–17.5%; *p* > 0.05); rate of dialysis (OR = 1.25; 95% CI 0.90–1.74; *p* > 0.05); rate of death (OR = 1.16; 95% CI 0.81–1.68; *p* > 0.05).
Dashti–Khavidaki S et al. 2013 [31] Iran	Cluster RCT 34/26 6	Hemodialysis (100.0%)	DM, HTN	Pharmacist provided nutrition consultation and motivational interviewing to patients, evaluated medication adherence and DRPs, educated about the disease, medications, lifestyle modification. Dose adjustment (1).	primary: difference in median HRQoL at the initiation and at the end of 6-month study (56.9; IQR 37.7–71.7 vs. 72.2; IQR 55.3–83.7; *p* = 0.001) in the intervention group; as well as in role−emotional, mental health, social functioning, and general health dimensions
Mateti UV et al. 2017 [32] India	RCT 78/75 12	Hemodialysis (100.0%)	CAD, CVD, CTD, DM, hypothyroidism, HTN	Pharmacists provided patient motivation and education about drugs, disease, lifestyle modifications, and diet, performed personal interview, and MRs. Validated pictogram−based information leaflets could be used (1).	primary ¶: change in HRQoL, e.g., SF−12 physical function (11.43; 95% CI 9.59−13.26; *p* < 0.0001) #, ESRD–kidney function (9.12; 95% CI 3.90−14.34; *p* < 0.0006) # at 12th month from baseline
Pai AB et al. 2009 [33] USA	RCT 61/46 24	Hemodialysis (100.0%)	DM, HTN	Nephrology−trained pharmacist conducted MRs, provided education, services optimizing drug therapy during rounds/formal reviews of the patients with the multidisciplinary health care team (1).	Total RQLP scores at year 1 (71 ± 34 vs. 88 ± 31; *p* = 0.03); Eating and Drinking score (4.4 ± 3.1 vs. 5.9 ± 3.3; *p* = 0.04); Physical Activities (30 ± 16.3 vs. 37 ± 13.6; *p* = 0.04); Leisure Time scores (5.9 ± 3.6 vs. 8.3 ± 3.4 *p* = 0.03).
Marouf BH et al. 2020 [34] Iraq	RCT 60/60 4	Hemodialysis (100.0%)	DM, HTN	Pharmacist created in−hospital guidelines for proper use of recombinant human erythropoietin, provided drug information on CKD−associated anemia to physicians and nurses, performed intervention at the physician, drug, patient, and hospital level (1).	primary: change in serum hemoglobin (0.91 g/dL; 95% CI 0.79–1.01; *p* < 0.0001) #, transferrin saturation (5.00%; 95% CI 4.77–5.23; *p* < 0.0001) #; ferritin (−90.70 ng/mL; 95% CI −113.60–(−67.80); *p* < 0.0001) # at 4th month from baseline; secondary: change in serum vitamin B12 (91.80 ng/mL; 95% CI 85.23–98.31; *p* < 0.0001) # and folate (1.39 ng/mL; 95% CI 1.15–1.63; *p* < 0.0001) # at 4th month from baseline.
van den Oever FJ et al. 2020 [35] the Netherlands	RCT 65/62 13	Intermittent, maintenance hemodialysis (100.0%)	Active malignancy, AF, CAD, HF, PVD, stroke/TIA	Pharmacist developed treatment algorithms for the dosing of DA and iron sucrose developed by pharmacist and provided dose advice (1).	primary: percentage in target range per patient for hemoglobin (6.8–7.4 mmol/L) (38.5; IQR 6.7–53.9 vs. 23.1; IQR 9.1–46.2%; *p* = 0.001); percentage of high hemoglobin levels (>8.1 mmol/L) (0.0; IQR 0.0–12.9 vs. 7.7; IQR 0.0–27.3%; *p* = 0.034); weekly dose of darbepoetin alfa (34.0; IQR 20.0–60.5 vs. 46.9; IQR 25.8–77.7 mcg; *p* = 0.020), iron sucrose dose (75; IQR 50–100 vs. 0.0; 0–100 mg/week; *p* < 0.001); mortality (RR = 0.6; 95% CI 0.34−1.03; *p* = 0.066).
Mateti UV et al. 2018 [36] India	RCT 78/75 12	Hemodialysis (100.0%)	CAD, CVD, CTD, DM, HTN, hypothyroidism, COPD	Pharmacists provided patient motivation and education about drugs, disease, lifestyle modifications, and diet, performed personal interview, and MRs. Validated pictogram−based information leaflets could be used (1).	primary ¶: change in SBP (−6.26mmHg; 95% CI −7.49–(−5.03; *p* < 0.0001) #; DBP (−3.56 mmHg; 95% CI −4.19–(−2.93); *p* < 0.0001) #; medication adherence (1.07; 95% CI 0.94–1.20; *p* < 0.001) #; hemoglobin (0.35 g/dL; 95% CI 0.22–0.48); *p* < 0.0001) # at 12th month from baseline
Qudah B et al. 2016 [37] Jordan	RCT 29/27 3	Hemodialysis (100.0%)	CV, DM, Dys	After obtaining home BP readings, pharmacist provided and discussed recommendations with the physician (acceptance or rejection). Educated and explained the goals for BP and daily weight gain (1).	primary: percentage of patients who reached weekly average home BP target of SBP < 135 mmHg and DBP < 85 mmHg (46% vs. 14.3; *p* = 0.02); change in average weekly home SBP (−14.40 mmHg; 95% CI −24.22–(−4.58); *p* = 0.004) # and DBP (−3.90 mmHg; 95% CI −10.02–2.22; *p* > 0.05) # at 3rd month from baseline
Pai AB et al. 2009 [38] USA	RCT 57/47 3	Hemodialysis (100.0%)	DM, HTN	Nephrology trained pharmacist conducted one−on−one patient interviews, generated a drug therapy profile; identified and addressed various DRPs through MRs, and provided patient education (1).	All−cause hospitalizations normalized per 1000 patient−days (1.8 ± 2.4 vs. 3.1 ± 3; *p* = 0.02), the cumulative time hospitalized (9.7 ± 14.7 vs. 15.5 ± 16.3 days; *p* = 0.06); mortality (RR = 1.03; 95% CI 0.54−1.98; *p* > 0.05)
Bessa AB et al. 2016 [39] Brazil	RCT 64/64 3	Kidney transplant (100.0%)		Pharmacist provided a predefined instructions given from day 3 to day 90 after kidney transplantation (1).	primary: %CV for tacrolimus concentrations (31.4% ± 12.3% vs. 32.5% ± 16.1%, *p* > 0.05); proportion of non−adherent patients at day 28 (17% vs. 26%, *p* > 0.05) and day 90 (27% vs. 25%, *p* > 0.05)
Chisholm M et al. 2011 [40] USA	RCT 12/12 12	Kidney transplant (100.0%)		Pharmacist conducted MRs, provided recommendations to the nephrologists, counselled patients about therapy, instructed on proper drug administration (1).	primary: compliance rate for immunosuppressive therapy (96.1 ± 4.7% vs. 81.6 ± 11.5%; *p* < 0.001) at the end of 1st year post−transplant; duration of compliance between the groups (*p* < 0.05).
Fleming JN et al. 2021 [41] USA	RCT 68/68 6	Kidney transplant (100.0%)	DM, HTN	Pharmacist conducted supplemental therapy monitoring and management, utilizing a smartphone−enabled mHealth app, integrated with risk−driven televisits and home−based BP. Conducted MRs at discharge, and provide recommendations to the patient (1).	secondary: the impact of the intervention on tacrolimus intrapatient variability over 12 months after randomization (*p* = 0.0133)
Gonzales H et al. 2021 [42] USA	RCT 68/68 12	Kidney transplant (100.0%)		Pharmacist monitored therapy via a mobile health−based application, integrated with risk−guided televisits and home−based BP and glucose measurements (1).	primary: pharmacist intervention types were medication reconciliation and patient education, followed by medication changes; secondary: 15% decrease in high−risk patients and a corresponding 15% increase in medium- or low-risk patients at 12th month from baseline
Gonzales H et al. 2021 [43] USA	RCT 68/68 12	Kidney transplant (100.0%)		Pharmacist monitored therapy via a mobile health−based application, integrated with risk−guided televisits and home−based BP and glucose measurements (1).	primary: change in medication errors (RR = 0.39; 95% CI 0.28–0.55; *p* < 0.001); secondary: rate of hospitalizations (RR = 0.46; 95% CI 0.27–0.77; *p* = 0.005)
Joost R et al. 2014 [44] Germany	quasi−exp 35/32 12	Kidney transplant (100.0%)		In addition to standard transplant training, pharmacist provided educational, behavioral and technical interventions, and consultations (transplant rejection, immunosuppressive drug actions and dosing, drug−drug interactions, common adverse effects, and adherence) (1).	primary: patient daily adherence, according to MEMS–percentage of days with the correct dosing of mycophenolate mofetil (91%, 95% CI 90.52–91.94 vs. 75%, 95% CI 74.57–76.09; *p* = 0.014) during the 1st year after transplantation; secondary: TA (95% ± 7.15 vs. 82% ± 20.2; *p* = 0.006); TiA (95% ± 7.88 vs. 94% ± 7.33; *p* > 0.05); PC (97% ± 7.33 vs. 90% ± 11.99; *p* = 0.008).

ACEI—Angiotensin-Converting Enzyme Inhibitor; ACR—Urine albumin/creatinine ratio, AF—Atrial Fibrillation; AKI—Acute Kidney Disease; ARB—Angiotensin Receptor Blocker; BP—Blood Pressure; CAD—Coronary Artery Disease; CCB–Calcium channel blocker; CKD—Chronic Kidney Disease; %CV—Coefficient of variation, calculated from 6 dose-corrected whole blood tacrolimus trough concentrations; CV—Cardiovascular Disease; CVD—Cerebrovascular Disease; DA—Darbepoetin; DM—Diabetes Mellitus; DRP—Drug Related Problem; DrugTEAM—collaborative multidisciplinary drug therapy evaluation and management service; Dys—Dyslipidemia; ESRD—End-Stage Renal Disease; HF—Heart Failure; HTN–Hypertension; HRQoL–Health Related Quality of Life; IQR—interquartile range; MEMS—Medication Event Monitoring System; MMAS—Morisky Medication Adherence Scale; MR—Medication Review; MTM—Medication Therapy Management program; OR—Odds Ratio; PC—Pill Count; PCP—Primary Care Provider; ProFiL—Training and Communication Network Program in Nephrology for Community Pharmacists; PVD—Peripheral Vascular Disease; RCT—Randomized Controlled Trial; RQLP—Renal Quality of Life Profile; TA—Taking Adherence, percentage of doses taken (bottle opening) in comparison to the total number of doses prescribed; TiA—Timing Adherence, percentage of doses taken within a 6-h interval (±3 h) around patients standard intake time; *—9 moth intervention; **—24 month intervention; ^—CKD or DM; #—differential change from baseline: intervention vs. usual care; ¶—according to data from academic hospitals patients; ¥—intervention/usual care; (1)—interventions provided by clinical pharmacists, (2)—interventions provided by community pharmacists.

They also conducted medication reviews and drug reconciliation (N = 15, 45.4%), gave instructions how to properly take medications, developed algorithms for the dosing of immunosuppressive agents or medication adjustment for kidney function (N = 11, 33.3%), focused on BP (tele)monitoring and therapy (N = 12, 36.4%), and provided recommendations for nephrologists/other physicians (N = 12, 36.4%). The follow-up periods ranged from 1 to 24 months (median nine months; IQR 3–12).

### 3.4. Data Synthesis and Analysis

The study results were tabulated and reported qualitatively. Due to a high degree of heterogeneity of the reported outcomes, the meta-analysis included the following data: change in systolic (SBP) and diastolic blood pressure (DBP) and/or proportion of patients with BP goal achieved, according to JNC-7 or JNC-8 inclusion, as well as change in eGFR and change in DRPs per patient. In addition, adherence was expressed as change in Morisky and Green medication adherence scale (MMAS), prescription refill, or a medication event monitoring system (MEMS), as described in Table 2.

#### 3.4.1. Meta-Analysis Results on Selected Clinical Outcomes and Adherence

The results are presented in Figure 4 and Figure 5. The analysis, through the random-effects model, found a statistically significant decrease in SBP (*p* < 0.0001) and DBP values (*p* < 0.0001) in patients that received interventions as compared to usual care (N = 11 studies). This effect remained relevant following further analyses in subgroups, which consisted of hemodialysis vs. remaining patients with renal failure. The impact of a variety of activities that were taken by pharmacists, in co-operation with other health-care providers, was significant (*p* = 0.0021) when considering the odds ratio for achieving target BP, i.e., below 140/90 mmHg during the follow-up period (N = 12 studies). Four studies examined differential change in eGFR from baseline; these yielded some effect (*p* = 0.072) for multidisciplinary intervention taken to improve chronic kidney disease.

In five studies, the analysis of dichotomous outcomes found that pharmacist-led interventions have an insignificant impact on improving medication adherence, as assessed by 4- and 6-MMAS score (*p* = 0.062). Similarly, the polled analysis of continuous data (differential change from the baseline) of 6- and 8-MMAS found no significant differences between usual care and intervention groups (N = 3 studies). Some effect was found when considering the number of DRPs per patient: at the end of some studies (N = 3 studies), the index was decreased from baseline in the intervention group compared to usual care (*p* = 0.053).

#### 3.4.2. Systematic Review of Other Outcomes

Two studies indicated a positive outcome, i.e., a decrease in cardiovascular risk (*p* < 0.001). In one study, this effect was associated with an improvement in the lipid profile (*p* = 0.004), SBP (*p* < 0.0001) or tobacco cessation (*p* = 0.04) by patients with CKD at stages 3 to 5.

Only four studies demonstrated a change in eGFR, from baseline to the end of the study, compared with controls up (see above). Other reported measures of renal function included protein (albumin)/creatinine ratio (ambiguous results) (N = 2 studies); change in serum hemoglobin (*p* < 0.0001) (N = 3 studies) and percentage in target range per patient for hemoglobin: 6.8–7.4 mmol/L (*p* = 0.001) (N = 1 study), transferrin saturation (*p* < 0.0001); ferritin (*p* < 0.0001) (N = 1 study); CKD progression, i.e., change of staging of CKD by eGFR criteria (ambiguous results) (N = 2 studies); proportion of patients with eGFR decline greater than 4 mL/min/1.73 m^2^ (*p* < 0.001) and composite of relative maximum change in creatinine, dialysis, and death at seven days after randomization (*p* > 0.05) (N = 1 study); median maximum relative change in creatinine concentrations (*p* > 0.05) (N = 1 study); rate of dialysis (*p* > 0.05) (N = 1 study). In three studies, intervention was found to influence glycated hemoglobin levels; in two studies, a significant increase was observed (*p* < 0.001).

There were seven papers that concerned the benefits of multidisciplinary care, including pharmacist-led interventions, on the medication adherence of patients with renal failure. In addition to the results from polled analysis of continuous or dichotomous data (see above), which revealed changes in MMAS score, one study reported better compliance for immunosuppressive therapy, at the end of the first after transplantation (*p* < 0.001), and duration of compliance between the groups (*p* < 0.05), while others indicated better patient daily adherence, expressed as the percentage of days with the correct dosing of the immunosuppressive agent, mycophenolate mofetil (*p* = 0.014), during the first year after transplantation (one study), as well as tacrolimus intrapatient variability (*p* = 0.013) (one study).

There were seven studies that reported drug prescription rates. Only one study demonstrated statistically significant increases in the prescription of relevant drugs. These included RAAS inhibitors (*p* = 0.06), calcium channel blockers (*p* < 0.0001), and diuretics (*p* < 0.0001). There were no significant differences in the prescription of NSAID (N = 1 study) or statins (N = 2 studies).

In addition to the continuous data regarding DRPs per patient, included in the meta-analysis (see above), five studies provided descriptive analyses of DRPs and types of recommendations that were proposed for patients receiving pharmacist-led interventions. In one study of the 182 patients reviewed, 42 recommendations were identified by the pharmacist; 85.7% recommendations were made on the basis of medications being dosed too high, and 14.3% were made owing to contraindication of use with current renal function. The most commonly identified medications requiring intervention were allopurinol (16.7%), spironolactone (14.3%), cetirizine (9.5%), tramadol (11.9%), and metformin (7.1%) [23]. In another trial, 108 DRPs were detected and managed by the clinical pharmacist, these concerned adding a new indicated drug to the existing medications (35.2%), discontinuation of unnecessary medications (22.2%), dose adjustments based on kidney failure (13.0%), changing route of drug administration (13.0%), correcting time of drug administration regarding food and interacting drugs (12.0%), and others [31]. Pai et al. (2009) reported the identification of 530 DRPs and their solution (N = 57 patients) [38]. Of these, 55% were related to renal osteodystrophy and cardiovascular therapy, and 25%, 21%, and 14% were categorized as medication record discrepancies, indication without a drug, and subtherapeutic dosage, respectively. In a study by Quintana-Bárcena et al. (2018), 897 DRPs were identified at baseline in 442 patients. Among these, 18.5% were associated with OTC medications not recommended for CKD, such as antacids or bisphosphonates, drugs requiring a dosage adjustment (9.6%), mainly anti-infectives, as well as nonadherence to drug therapy (30.2%), particularly antihypertensives and uncontrolled blood pressure [25]. Song et al. (2021) identified 182 DRPs in 50 patients. The majority of the DRPs concerned indication without prescription (35.7%), inappropriate dosage/administration (31.9%), no indications (11.0%), inappropriate drug prescription (6.6%) or ADRs (11, 6.0%), problems with drugs for the management of such comorbidities as mineral bone disorder (19.8%), hypertension/cardiovascular diseases (18.7%), and anemia (13.2%) [26]. The majority of the DRPs concerned indication without prescription (35.7%), inappropriate dosage/administration (31.9%), no indications (11.0%), inappropriate drug prescription (6.6%) or ADRs (11, 6.0%), problems with drugs for the management of such comorbidities as mineral bone disorder (19.8%), hypertension/cardiovascular diseases (18.7%), and anemia (13.2%). In two studies, the medical team demonstrated high acceptance for the pharmacist-driven proposed changes, i.e., from 81.9% to 100% [26,38].

Five studies reported the impact of pharmacist-led interventions on the health related quality of life (HRQoL) of CKD patients. Due to different scales (SF-12, SF-36, RQLP, EQ-5D-5L) and the heterogeneity of results obtained in particular protocols, no pooled meta-analysis was performed. Ambiguous results regarding HRQoL improvement were obtained, from participants, from the Intervention group as compared to Usual care; N = 4 studies demonstrated a positive impact on the outcome.

In some protocols, an outcome of all-cause hospitalizations was reported. It was expressed as odds ratio (*p* > 0.05) (N = 2 studies) and as risk ratio (*p* = 0.005) (N = 1 study); the composite outcome of first acute care events (hospitalization and emergency department and urgent care center visits) (*p* > 0.05) (N = 1 study) was assessed. Mortality expressed as risk ratio (odds ratio) (N = 5 studies) was insignificantly decreased compared to the control groups.

## 4. Discussion

The present survey provides an updated review and polled analysis of RCTs of the effectiveness of different pharmacist-led interventions, mostly by clinical pharmacists, for CKD against a range of outcomes. There were 33 studies included in this review, and the data from 19 protocols were included in further quantitative analyses. The polled analysis of data from fifteen studies indicated improved blood pressure values in the short term.

A previous meta-analysis by Nakanishi et al. (2020), including six studies, found pharmacist-led interventions, especially those concerning home-based BP telemonitoring, to be significantly superior to standard care, expressed as odds ratio. The follow-up period ranged from three to eighteen months, and interventions taken by pharmacists included home BP telemonitoring conduction in two out of fifteen studies that reported the BP values [5]. This paper is also consistent with the present findings. The novelty of the current meta-analysis is that such a benefit was calculated for both dichotomous data (odds ratio) and continuous data (mean differential change at the end of the study from the baseline) for the intervention group, which generally received other interventions than BP telemonitoring, compared to standard care. The further subgroup analysis also indicated that clinical pharmacist-led interventions resulted in benefits from both ESRD patients receiving hemodialysis and those with renal failure but not needing dialysis. The analyses demonstrated absence of publication bias across these studies, as well.

A systematic review by Kamath et al. (2020), including 22 RCTs and non-randomized studies, found improvement of hypertension management in CKD patients. Most of the included interventions were performed by primary care physicians; only some described collaborations between community pharmacists and other health care providers. The authors emphasize the potential of clinician-facing interventions to improve BP outcomes in primary care [45].

The relative effectiveness of CKD management strategies has been examined in recent systematic reviews of both prospective and retrospective studies based on models of multidisciplinary care; these studies were performed by various specialists, including nephrologists, nurses, surgeons, general practitioners, pharmacists, psychotherapists, social workers, and nutritionists. The analysis based on qualitative evidence, but not pooled quantitative evidence, found that such interventions resulted in significant reductions in patient mortality, the risk of renal replacement therapy, and the need for emergent dialysis, as well as a decrease in annual medical costs. Moreover, the systematic reviews showed that the pharmacist contributions had a positive effect on clinical outcomes, such as the levels of hemoglobin, parathyroid hormone, and calcium, as well as creatinine clearance or eGFR [1,46,47].

The present meta-analysis of differential change in eGFR from baseline, as analyzed in four studies, yielded some benefit, e.g., indicating slower progression of renal failure (*p* = 0.072) associated with multidisciplinary interventions, including a medication dose adjustment for kidney function. However, the follow-up ranged between 3 to 12 months, and the heterogeneity between studies was high. Such measures of renal function as protein (albumin)/creatinine ratio, change in serum hemoglobin or creatinine concentrations, transferrin saturation and ferritin, and dialysis rate were measured in two studies (either one or the other); most of these were addressed to hemodialysis patients.

The type of intervention had no significant influence on improving medication adherence in patients with CKD. In more than 70% of studies, pharmacist-led patient intervention included such activities as patient-oriented motivation and education about medications (e.g., goals of therapy, potential side effects, and/or contraindications), disease, lifestyle changes, and nutrition. However, only a few RCTs provided data for resultant medication adherence; the measures were described as 4-, 6-, or 8-MMAS score. The polled analysis of dichotomous data revealed some increase in odds ratio for percentage of patients with good adherence (*p* = 0.062). Nevertheless, medication adherence among patients with CKD seems to be a continual challenge, despite various educational interventions. In addition, other systematic reviews found the various educational methods to have an unclear effect on medication adherence in other chronic diseases, e.g., hypertension, heart failure, or diabetes [48,49,50,51]. In a recent meta-analysis of nine studies, a variety of outcome measures was considered, such as adherence, pill count, MMAS-4 and 8, prescription refill, and medication possession ratio (MPR). The results show that educational interventions targeted at hypertensive patients were minimally effective in improving medication adherence as compared to standard care at healthcare facilities. The authors conclude that an increase in the frequency of communications with a health care professional may improve habits by reducing forgetfulness, which is one of the factors hindering medication adherence [7]. Interestingly, the pharmacist-driven interventions that were addressed to patients after kidney transplantation, including instructions on proper medication use and/or dosage adjustments, did result in a significant improvement in compliance rate for immunosuppressive therapy and a decrease in intrapatient variability. These are well-known predictors of risk of allograft loss [52]. However, the sample sizes were too low to make solid conclusions concerning rejection rates and compliance.

In most of the studies given above, clinical pharmacists worked as part of multidisciplinary care teams with nephrologists (or other health care professionals), nurses, and nutritionists. Undoubtedly, clinical pharmacy services have the potential to contribute significantly to multidisciplinary teams, providing safe, effective, and inexpensive care [47]. Clinical pharmacists can play important roles in the multidisciplinary care of CKD patients, such as managing anemia, renal mineral bone disease, and hypertension, as well as more general services, as proposed by Mason et al. (2010) [53]. An emerging role is the opportunity to prescribe and modify medication therapy and dosing, or dealing with drug–drug interactions, which has now been implemented into practice in some countries, including the United Kingdom, New Zealand and United States [54]. Most of the clinical trials included in the current analysis were performed in the US (17/33), and this seems to reflect the growing importance of clinical pharmacists in the US health care system. In contrast, only individual studies came from Brazil, India, Iran, Iraq, or South Korea, where the numbers of pharmacists supporting multidisciplinary teams are limited. In these locations, pharmaceutical services mainly include dispensing of medicines, patient education about their usage, management with psychotropic drugs, or control of medicine storage conditions. The impact of clinical pharmacy services on care of inpatients with CKD has yet to be established [26,31,32,34,36,55]. Therefore the proposed pharmacist-led interventions in these trials could deviate from real-world situations. A key limitation of the present survey is its great homogeneity with regard to geographic location, health care system organization, and clinical setting.

In the present systematic review, pharmacists provided patient motivation and education in most studies; they undertook medication chart review to identify DRPs relatively frequently (45.4%). In advanced health systems, in nephrology centers, pharmacists can specifically focus on the control of possible negative interactions between drugs and determine the appropriateness of dosages according to the residual renal filtrate of patients. However, such activities were proposed only in a few studies (15.2%), and they were mostly performed by nephrology-trained pharmacists [22,27,29,35,38].

In protocols where pharmacists performed MRs for CKD patients, some benefit was found for decreasing the number of DRPs per patient; however, this index was reported in only three studies (*p* = 0.053). Similarly, only some studies performed a detailed qualitative analysis of DRPs identified by pharmacists in the intervention groups; subsequently, the pharmacists gave a number of recommendations to physicians: these concerned the medication/dosage adjustments based on kidney failure, adding a new indicated drug to the medication regimen, withdrawal of OTC medications that were not recommended for CKD, or dealing with adverse drug reactions. Interestingly, only in two studies the acceptance rate by the medical team, for the pharmacist-driven proposed changes, was high [26,38]; some studies failed to provide such data. Similarly, a number of recommendations were proposed by pharmacists according to particular cases, such as patients with a risk of CV, those using RAAS inhibitors (albuminuria), diuretics, calcium channel blockers or statins, as well as those who had to avoid nonsteroidal anti-inflammatory drugs (NSAIDs) or allopurinol, or those requiring a dose adjustment for metformin or tramadol. However, the outcomes concerning the improved rates for medication use were rarely reported, and these results were unclear. Those few studies that reported the impact of pharmacist-led interventions on the quality of life, another important measure for CKD patients, demonstrated high heterogeneity and variety of scales; nevertheless, four indicated that the performed activities demonstrated benefits in relation to role-emotional, mental health, social functioning, and/or general health dimensions.

The present study was restricted to RCTs to limit the potential for bias. Moreover, in most studies, the percentage of patients with CKD was almost 100%. If possible, for the purposes of meta-analysis, the overall effect size was calculated as mean differential change from baseline: intervention vs. usual care for continuous data that were measured at the beginning and at the end of study for patients from each group. To improve the presentation of the results from the various studies included in the systematic review, the effect sizes were also calculated as odds ratio, for dichotomous data, or as the mean difference at the follow-up between the intervention and usual care groups for continuous data.

Despite these strengths, the study does have limitations. Firstly, the included studies demonstrated high heterogeneity with regard to participant characteristics, intervention type (contact time with the patient, duration and type of intervention, etc.), follow-up, and type of measure, e.g., 4-, 6-, 8-MMAS, prescription refill, rate of compliance, or MEMS–for medication adherence, or SF-12, SF-36, RQLP, EQ-5D-5L–for health-related quality of life. Unfortunately, it was not possible to explicitly address this heterogeneity, as the included studies often had insufficient information to undertake subgroup analyses based on clinical (e.g., cardiovascular, renal) or non-clinical aspects (e.g., medication adherence), patient condition (e.g., severity of renal failure, co-morbidities), or type of performed intervention or length (e.g., 9 out of 33 studies lasted 1 to 3 months). In addition, the studied material was subject to a high risk of bias: only 12.1% studies were judged as low risk.

## 5. Conclusions

Despite the low-to-moderate quality of evidence and relatively short follow-up, current findings suggest that multidimensional pharmacist-driven interventions within multidisciplinary teams have a positive influence on some clinical outcomes, such as blood pressure, CV risk, and renal progression, and they support adherence to medication among individuals with renal failure. Future research should focus more on medication reviews and assessments of medication usage, as well as examining the effectiveness of recommendations, given by pharmacists, on adding new drugs to medication regimens, discontinuing unnecessary medications, or dose adjustments based on kidney failure. The long-term outcomes should be tested in real-world studies, encompassing patient populations with diverse characteristics and healthcare interventions. The obtained findings could, then, be potentially generalizable to the broader population of patients with renal failure.

## Figures and Tables

**Figure 1 ijerph-19-11170-f001:**
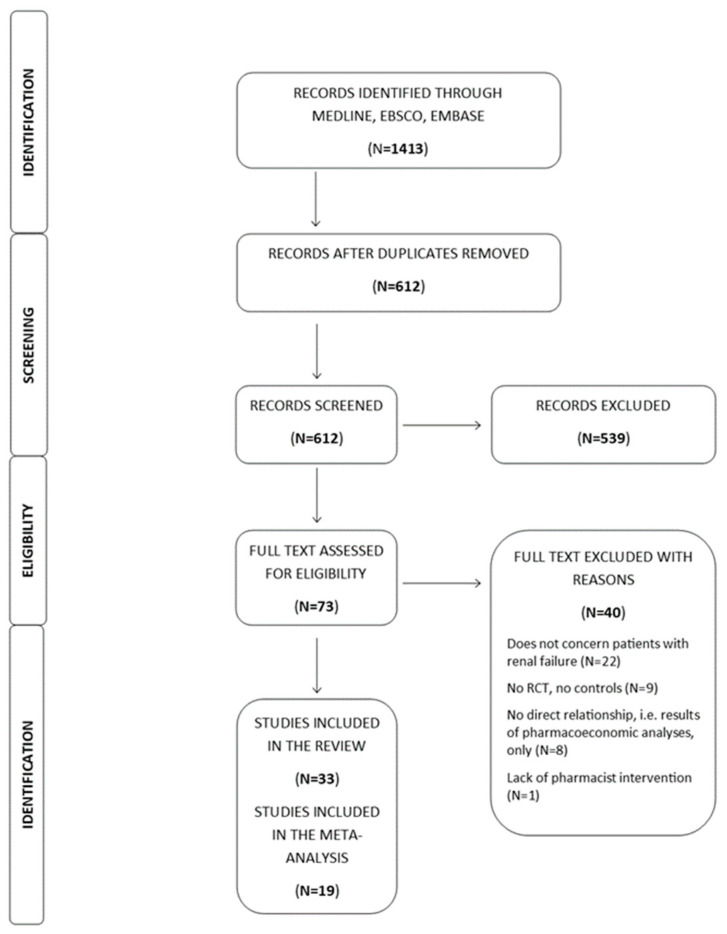
Flow diagram of the search study.

**Figure 2 ijerph-19-11170-f002:**
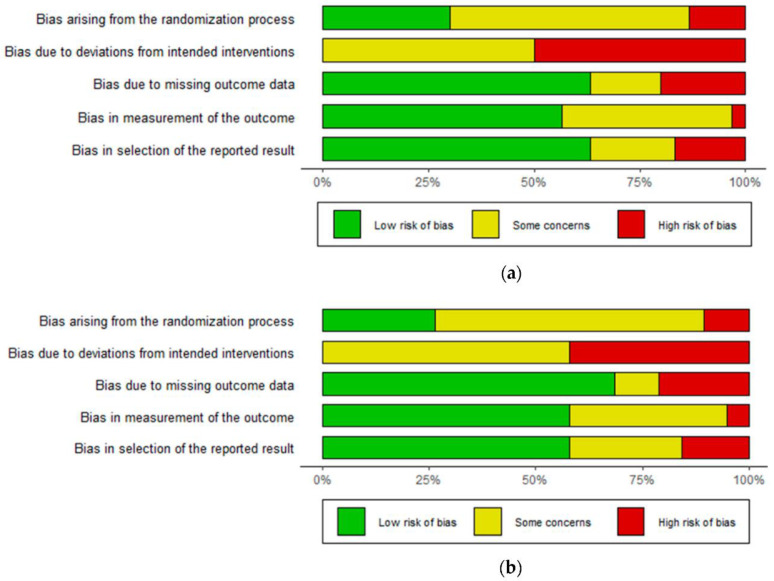
Summary of risk of bias. All studies, N = 33 (**a**), studies included into meta-analysis, N = 19 (**b**).

**Figure 3 ijerph-19-11170-f003:**
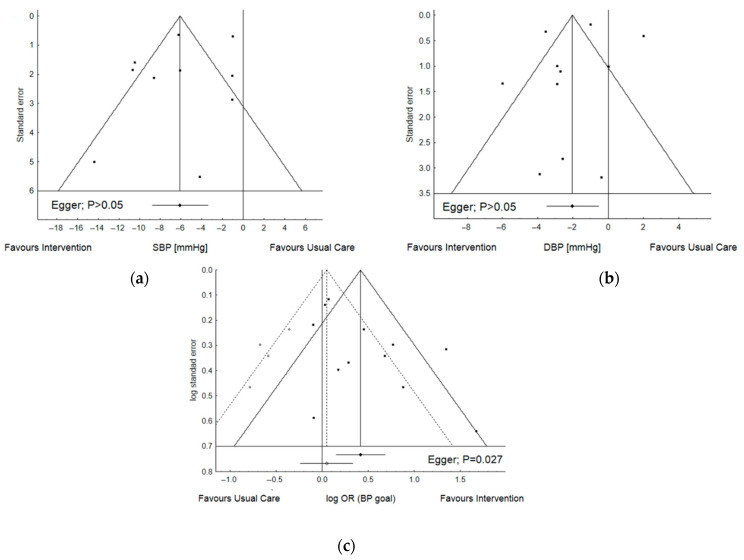
Funnel plot of the studies included in the meta-analysis of differential change from baseline in SBP (**a**) and DBP (**b**), as well as odds ratio for achieving BP goal (**c**). Graphs show the distribution of published study outcomes (filled squares) vs. unpublished outcomes (open circles) estimated by Trim and Fill analysis. The dashed line represents the mean and 95% CI with the added, potentially unpublished, studies, and the solid line represents published studies included in meta-analysis. Vertical dashed line represents the global estimate of efficacy. Overall effect size D = −5.86 [−8.31, −3.41]—0 potentially missing studies were added (**a**); D = −3.30 [−3.88, −2.72]—0 potentially missing studies were added (**b**); OR = 1.52 [1.16, 1.98] vs. 1.05 [0.79, 1.40]—6 potentially missing studies were added (**c**).

**Figure 4 ijerph-19-11170-f004:**
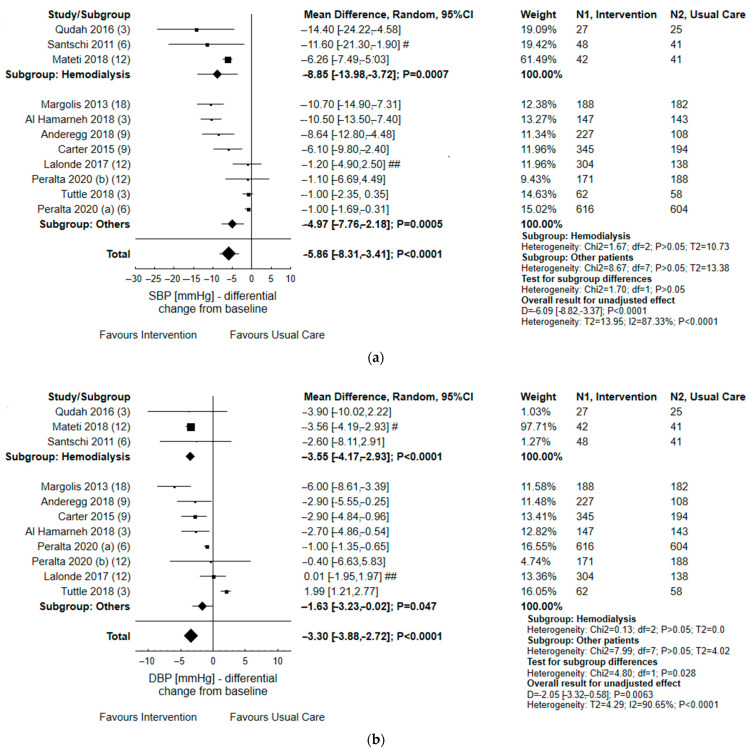
Forest plot of comparison: pharmacist contribution to comprehensive intervention vs. usual care for continuous and dichotomous data concerning clinical outcomes [12,13,14,15,17,18,19,20,21,22,24,27,28,29,36,37]. Differential change from baseline in SBP (**a**) and DBP (**b**); patients with an achieved BP goal (**c**); differential change from baseline in eGFR (**d**); #—adjusted for sex, baseline systolic BP, smoking status, body-mass index, and cluster characteristics (mean pharmacist year since graduation); ##—adjusted for the clinical variable at baseline (eGFR), interaction between study group and clinical variable at baseline, and patient’s age, sex, highest level of education, and eGFR, as well as pharmacist being an associate clinician and receiving remuneration for pharmaceutical opinions; BP goal was defined as <140/90 mmHg, except for: &—140/90 mmHg for non-proteinuric CKD and <130/80 for proteinuric CKD; ^—<140/90 mmHg for non-hypertensive patients, and <130/80 mmHg for hypertensive patients; $—<140/90 mmHg for subjects with neither condition, and <130/80 mmHg for DM/CKD patients *—<130/80 mmHg; months of follow-up were shown in brackets.

**Figure 5 ijerph-19-11170-f005:**
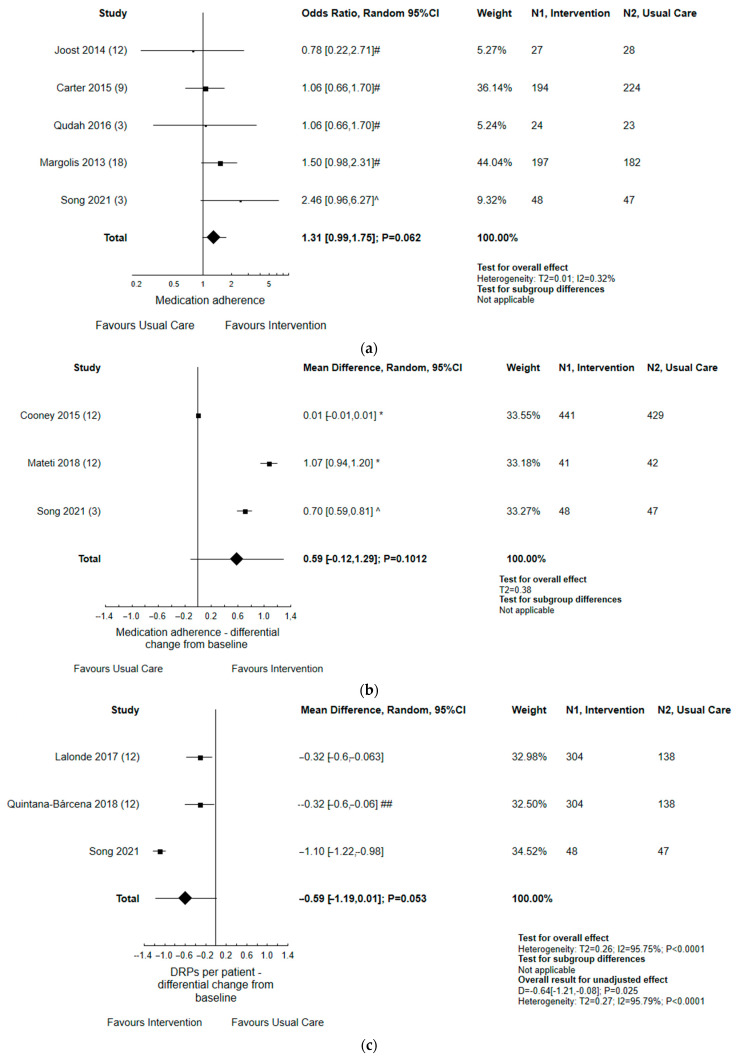
Forest plot of comparison: pharmacist contribution to comprehensive intervention vs. usual care for continuous and dichotomous data concerning medication adherence (**a**,**b**); DRPs per patient (**c**); #—patients with good adherence, defined as 0 (1), according to 4-item Morisky Medication Adherence Scale (MMAS) score; ^—patients with increased 6-item MMAS score; *—patients with increased 8-item MMAS score; ##—adjusted for the number of drug related problems (DRPs) at baseline, the interaction between study group and number of DRPs at baseline, and for patient’s age, sex, highest level of education, and eGFR, as well as for pharmacists’ being an associate clinician and receiving remuneration for pharmaceutical opinions; months of follow-up were shown in brackets [13,15,19,24,25,26,36,44].

**Table 2 ijerph-19-11170-t002:** Adherence measurement according to the reviewed protocols.

Measurement of Patient Adherence	Description
Compliance Rate (CR)	CR is calculated according to the formula: [number of medication agent doses filled by pharmacy/number of doses prescribed per time period] × 100%; with 80% as a minimum threshold [40].
Medication Event Monitoring System (MEMS)	An electronic medication bottle cap that records whenever the bottle is opened. Adherence is calculated according to the formula: [number of times bottle is opened/number of pills prescribed] × 100% [44].
Medication non-adherence	Patient takes less than 80% of prescribed doses or too many medication [24].
Morisky Medication Adherence Scale-4 (MMAS-4)	A series of four closed questions, each question that is answered with a ‘No’ receives a score of 1. The possible scoring range is 0 to 4. A score of 4 indicates high adherence, a score of from 2 to 3–medium adherence, and a score of less than 2–means poor patient adherence [13,37].
Morisky Medication Adherence Scale-4 (MMAS-4)-modified	MMAS-4 scale was modified for BP medications [15,44].
Morisky Medication Adherence Scale-6 (MMAS-6)	A series of six closed questions, 2 new questions were added to MMAS-4 scale to create the Modified Morisky Adherence Scale (MMAS-6). For the motivation domain (3 questions), a scoring range is from 0 to 3; a total score of 0 to 1 indicates low motivation and if the score is >1, the motivation domain is scored as high. For the knowledge domain (3 questions), a scoring range is from 0 to 3; a total score of 0 to 1 indicates low knowledge; if the score is >1, the knowledge domain is scored as high [26].
Morisky Medication Adherence Scale-8 (MMAS-8)	A series of eight closed questions, 4 new questions were added to MMAS-4 scale to create the Modified Morisky Adherence Scale (MMAS-8); each question that is answered with a ‘No’ receives a score of 1. A score of eight indicates high adherence; a score of six to seven indicates medium adherence and a score of less than six indicates poor adherence [19,36].
Basel Assessment of Adherence to Immunosuppressive Medication Scale (BAASIS) questionnaire	A series of six closed questions within three domains: implementation (items 1a, 1b, 2, 3–all items start with a ‘Yes’/‘No’ question; for items 1a, 1b and 2, if patient answers ‘Yes’, this is followed by five response categories to document the frequency of implementation problems, i.e., once, twice, etc.), initiation (item 5 with a ‘Yes’/‘No’ answer) and persistence (item 4 with a ‘Yes’/‘No’ answer). Any ‘Yes’ on any of items 1a, 1b, 2 or 3 indicates an issue with implementation. ‘Yes’ on item 4 indicates non-persistence of immunosuppressive medication use [39].

## Data Availability

Author can confirm that all relevant data are included in the article and–on request from the corresponding author.

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
