# Peer review of "The Effectiveness of Pharmacist Interventions in the Management of Patient with Renal Failure: A Systematic Review and Meta-Analysis"

_ijerph, 2022, doi:10.3390/ijerph191811170_

Round 1
Reviewer 1 Report
We have read with much attention and interest the review -metanalysis "The Effectiveness of Pharmacist Interventions in the Management of Patient with Renal Failure: A Systematic Review and 3 Meta-Analysis."
The paper reviews a large number of papers concerning the role of pharmacists in improving outcomes such as blood pressure control, progression of renal failure, and appropriateness of drug administration in dialysis and transplant patients. Most of the papers reviewed are interesting and valuable, but some might not have been included in the review because of the short period of observation. This would have greatly streamlined the text, which as organized is too long and somewhat redundant, making it difficult to read and focus on outcomes.
1. Tab.1 is too large (9 pages) and probably positioned in a way that does not facilitate the logical thread of the work.
2. Probably grouping the studies considered according to the main objective or prevalent population they target would facilitate understanding of the paper.
3. The discussion does not include a comparison with other reviews produced on the topic in recent years
4. The studies reviewed all belong to particular geographical areas ( Asia and North America predominantly). The role of pharmacists should be framed within the organizational reality of the country in which they operate.
5. The results obtained and presented in the various studies should be weighed against the health care organization and the possible interaction between pharmacists and patients and nephrologists in real-life situations and randomized clinical trials.
6. In advanced health systems, which offer universal health care, many of the roles of pharmacists proposed in the studies are mainly conducted by nephrological centers through their operators, while the role of pharmacists should be specifically addressed to the control of possible negative interactions between drugs and the appropriateness of dosages according to the residual renal filtrate of patients. These considerations have to be discussed in the text.
Reviewer 2 Report
The author has done a systematic review and meta- analysis of the The Effectiveness of Pharmacist Interventions in the Management of Patient with Renal Failure.
However the study is focused on management of hypertension. Hence the title needs to be changed to include hypertension instead of renal failure.
Overall well written.
Reviewer 3 Report
This is a very meaningful manuscript, the study indicates that multidimensional interventions taken by pharmacists within multidisciplinary teams are important for improving some clinical outcomes, such as blood pressure, risk of cardiovascular diseases and renal progression, and improve non-adherence to medication among individuals with renal failure. I recommend the acception of this manuscript for publication after minor revision with English language modification, refs updates.
Round 2
Reviewer 1 Report
The author has accepted the review's suggestions, quite completely. The paper, especially in the discussion, has become more attractive, as also the modification in table 1 have made it more easily accessible.